# Toward Approaches to Scalability in 3D Human Pose Estimation

**Jun-Hee Kim**     **Seong-Whan Lee**\*
Dept. of Artificial Intelligence, Korea University, Seoul 02841, Republic of Korea
{jh__kim, sw.lee}@korea.ac.kr

## Abstract

In the field of 3D Human Pose Estimation (HPE), scalability and generalization across diverse real-world scenarios remain significant challenges. This paper addresses two key bottlenecks to scalability: limited data diversity caused by *'popularity bias'* and increased *'one-to-many'* depth ambiguity arising from greater pose diversity. We introduce the Biomechanical Pose Generator (BPG), which leverages biomechanical principles, specifically the normal range of motion, to autonomously generate a wide array of plausible 3D poses without relying on a source dataset, thus overcoming the restrictions of popularity bias. To address depth ambiguity, we propose the Binary Depth Coordinates (BDC), which simplifies depth estimation into a binary classification of joint positions (front or back). This method decomposes a 3D pose into three core elements—2D pose, bone length, and binary depth decision—substantially reducing depth ambiguity and enhancing model robustness and accuracy, particularly in complex poses. Our results demonstrate that these approaches increase the diversity and volume of pose data while consistently achieving performance gains, even amid the complexities introduced by increased pose diversity.

## 1 Introduction

3D Human Pose Estimation (HPE) is essential for determining the 3D locations of human joints in images or videos. It plays a pivotal role in applications such as person re-identification [1], action recognition [2, 3, 4], human mesh recovery [5, 6], and virtual reality [7, 8]. As the importance of this technology grows, there is an increasing demand for more accurate and universally applicable models. Addressing the key challenges that limit the scalability of this field is therefore crucial.

In this paper, we identify two primary bottlenecks impeding the scalability of 3D HPE:

1. **Difficulty in Collecting Diverse Datasets**: Annotated 3D datasets for 3D HPE are typically collected in controlled indoor environments with limited motions performed by a few individuals. This results in a 'popularity bias'[9], where the data primarily reflects a small subset of human activities and demographics, thus restricting the diversity needed for effective model performance in various real-world scenarios. This lack of diversity means that models trained on such data may fail to generalize to unseen poses and environments.

2. **Exacerbation of the 'One-to-Many' Depth Ambiguity Problem**: Depth ambiguity[10, 11, 12] arises when a single 2D pose can correspond to multiple 3D interpretations. Increased pose diversity in 3D HPE intensifies this issue, complicating the depth estimation process. This ambiguity significantly impacts the accuracy of 3D pose estimation, as the model must choose from several plausible 3D configurations for a given 2D input, often leading to incorrect depth predictions.

---

\*Corresponding author

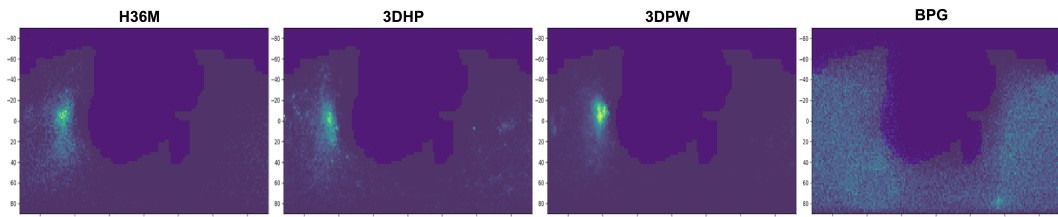

Figure 1: Spherical coordinate distribution for the vector connecting the right shoulder to the right elbow across datasets: H36M [20], 3DHP [21], 3DPW [22], and BPG, with valid regions highlighted in grey (taken from [23]).

To address the first bottleneck, pose augmentation methods for 3D HPE [13, 14, 15, 16] have been proposed. These methods leverage knowledge of labeled (source) data to enhance the generalization capability of pose estimators on unseen data. However, relying solely on source data does not fully overcome the inherent limitations, as the derived knowledge also carries the same 'popularity bias,' limiting its utility outside laboratory environments. Therefore, previous methods cannot guarantee the generation of out-of-source poses that are crucial for effective real-world applications.

To overcome these limitations, we propose a Biomechanical Pose Generator (BPG) that does not rely on source data but utilizes biomechanical knowledge. The concept of 'Normal Range of Motion (NROM)'[17] is commonly used in the medical field to describe the standard limits within which a joint can move without discomfort or injury. Our method applies the NROM constraint to the forward kinematic function[18], enabling the autonomous generation of a wide range of plausible 3D poses. This approach enhances the diversity and reliability of pose data.

Furthermore, recognizing that adult human proportions are generally consistent [19], we allow slight variations in proportions to accommodate a broader range of body types. Consequently, BPG leverages biomechanical knowledge to generate a broader spectrum of feasible 3D poses, thereby avoiding the popularity bias inherent in existing datasets, as demonstrated in Figure 1. This methodology provides the essential flexibility and adaptability required for accurately modeling and predicting the diverse range of human poses and activities in real-world environments.

To mitigate the one-to-many problem in 3D HPE, we developed the Binary Depth Coordinates (BDC) that reconceptualizes depth estimation as a binary classification task. Instead of using traditional continuous depth coordinates, we limit each joint's depth to two options: front or back. This simplification is based on geometric principles; once the length of connected bones is determined, two possible depth positions (front or back) can be calculated based on the joint positions. This approach allows us to decompose each 3D pose into three main components: 2D pose, bone length, and binary depth decision. This decomposition effectively reduces depth ambiguity and significantly enhances the model's accuracy and robustness for complex poses.

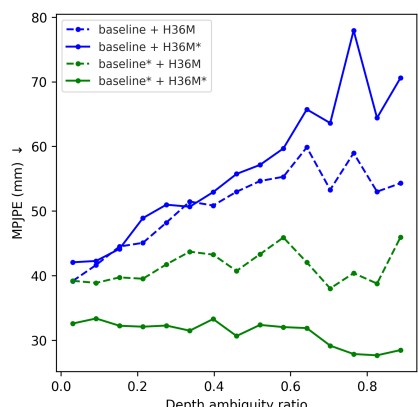

Figure 2: The MPJPE (in mm) across 3D HPE models against the depth ambiguity ratio is detailed in Sec.3.2. The baseline* employs BDC on the baseline model[24], and H36M* combines source data with additional data synthesized by BPG, expanding the total dataset to ten times the H36M training size.

As demonstrated in Figure 2, poses with significant depth ambiguity have shown poorer performance in the baseline model, with increased data volumes exacerbating this issue. However, when employing the BDC, performance remains consistent regardless of ambiguity, and notably, poses with greater depth ambiguity exhibit improved performance as the amount of data increases.

This paper aims to advance the development of scalable solutions in 3D HPE by addressing these issues, ultimately contributing to the creation of more accurate and adaptable technologies. By combining our data generation method with a new approach to resolving depth ambiguity, we aim to establish a foundation for future research and applications in this dynamic field.

## 2    Biomechanical Pose Generator

To address the limitations of current data collection methods, we propose the Biomechanical Pose Generator (BPG) that does not rely on source data but utilizes biomechanical knowledge. The concept of 'Normal Range of Motion (NROM)' is commonly used in the medical field to describe the standard limits within which a joint can move without discomfort or injury. Our method applies the NROM constraint to the forward kinematic function, enabling the autonomous generation of a wide range of plausible 3D poses. This approach enhances the diversity and reliability of pose data. Recognizing that adult human proportions are generally consistent, we allow slight variations in proportions to accommodate a broader range of body types. Consequently, BPG leverages biomechanical knowledge to generate a broader spectrum of feasible 3D poses, thereby avoiding the popularity bias inherent in existing datasets. This methodology provides the essential flexibility and adaptability required for accurately modeling and predicting the diverse range of human poses and activities in real-world environments.

**Osteo-kinematic Model.** The human body performs complex motions constrained by biomechanical principles. To capture these constraints, we utilize an osteo-kinematic model similar to [18] for generating realistic 3D human poses. This model is based on the biomechanical structure of the human body, including bones, joints, and their respective degrees of freedom (DOF). In the osteo-kinematic model, bones can rotate at connected joints, and each joint is assigned a specific number of DOF to represent natural human-like movements. For instance, elbows and knees each have one degree of rotational freedom (1-DOF), while shoulders, hips, neck, and pelvis each have three degrees of rotational freedom (3-DOF). Unlike typical osteo-kinematic models that use fixed bone lengths, our model allows for adjustments in human body proportions to generate poses for a diverse range of subjects. The model includes 17 joints, 16 bone lengths, and 25 angle parameters. This detailed configuration ensures that the generated poses adhere to the natural movement constraints of the human body while allowing for variability in body types. Figure 3 illustrates the joints and degrees of freedom in the osteo-kinematic model.

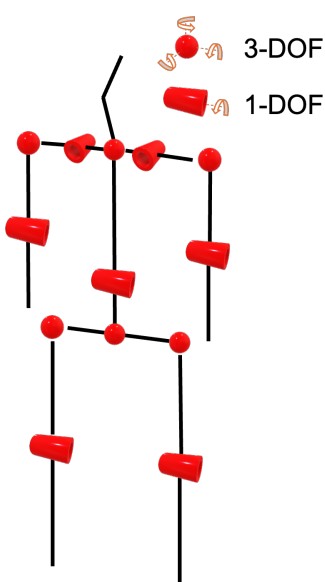

Figure 3: Osteo-kinematic Model with Joints and Degrees of Freedom.

**Postural Noise.** The osteo-kinematic model uses 41 parameters (lengths and angles) to create a pose. Unconstrained random sampling of parameters may generate unnatural 3D poses, such as a backward-bent arm or leg, which do not accurately represent natural human postures. To address this, we set boundaries using two predetermined human features: a NROM [17] and human limb bone length ratios [19]. The range of motion (ROM) refers to the extent to which bones can move through a complete spectrum of movements. Although each individual's ROM varies, each joint possesses a NROM representing healthy human movement. NROM encompasses three different axes of rotation, considering the movement direction of the joint. By aligning these axes with the DOF of the osteo-kinematic model, we define an interval $[\theta_i^{lower}, \theta_i^{upper}]$ for each $i^{th}$ angle parameter. Hence, these intervals allow a self-created pose to enable natural human motion. Different people exhibit distinct bone lengths. However, the proportions of the human body remain relatively consistency for most individuals. This constancy permits us to set a mean value for human limb length ratios $\bar{l}_j$ for the $j^{th}$ bone. To obtain 3D human data with various ratios, we establish a ratio range $[(1-\alpha)\bar{l}_j, (1+\alpha)\bar{l}_j]$ for each $j^{th}$ bone parameter, where $\alpha \in (0, 1)$ is a predetermined parameter. Using these intervals, we randomly obtain the bone parameter set $\boldsymbol{l} = \{l_b\}_{j=1}^B$ and the angle parameter set $\boldsymbol{\theta} = \{\theta_i\}_{i=1}^P$, where $B$ is the number of bones, $P$ is the number of angle parameters, $l_j \sim \mathcal{U}_{[(1-\alpha)*\bar{l}_j,(1+\alpha)*\bar{l}_j]}$ represents the bone parameter of the $j^{th}$ bone, and $\theta_i \sim \mathcal{U}_{[\theta_i^{lower},\theta_i^{upper}]}$ represents the $i^{th}$ angle parameter. Finally, we define these two sets as the postural noise $\mathbf{N}_{pose} = \{\boldsymbol{l}, \boldsymbol{\theta}\}$, which can be acquired randomly within boundaries representing the characteristics of ordinary individuals.

**Forward Kinematic Function.** Following [25], we use the forward kinematic function $\mathbf{T}$ to map postural noise to joint positions:

$$\mathbf{T}(\mathbf{N}_{pose}) = \mathcal{P}^{3D}, \tag{1}$$

where $\mathcal{P}^{3D} \in \mathbb{R}^{J \times 3}$ represents the synthesized 3D pose (see Appendix C.1). Each joint undergoes a local coordinate transformation based on its parameters. The global coordinates of each joint are calculated by multiplying a series of rigid transformation matrices obtained via the parameters from the root point to itself.

**Pose Confidence.** Although our method utilizes the NROM of individual joints to create realistic human poses, the relative positions of all bones must be considered to ensure the plausibility of the pose. For instance, disregarding the bone position can result in a physical intersection, such as legs crossing or hands intersecting with the torso. For this reason, we calculate the bone distance $d_{n,m}$ between the $n^{th}$ and $m^{th}$ bones. We use the indicator function of the interval $[\bar{d}_{n,m}, \infty)$, where $\bar{d}_{n,m}$ is the pre-calculated relative bone radius [19]. The pose confidence $C$ for $\mathcal{P}^{3D}$ is given by

$$C = \prod_{m=n+1}^{B} \prod_{n=1}^{B-1} \mathbb{1}_{[\bar{d}_{n,m}, \infty)}(d_{n,m}). \tag{2}$$

If any bones get abnormally close, the confidence value becomes 0; otherwise, it remains 1. This measure allows us to avoid poses that are anatomically possible but physically impossible. We eliminate poses with a value of 0 using the pose confidence $C$ and only use poses with a value of 1.

**Projection.** We project the generated 3D poses into 2D poses using camera parameters, including a rotation matrix, translation vector, focal length, and principal point. Without predetermined camera settings, we generate these parameters within our framework. We sample a rotation matrix $R \in \mathbb{R}^{3 \times 3}$ from a uniform distribution on all 3D rotation matrices and the translation vector $T \in \mathbb{R}^3$ from a uniform distribution on the interval $[T^{lower}, T^{upper}]$ that sets a camera location to avoid abnormally close or far positions. Although other methods use a limited rotation distribution, we use a comprehensive distribution to fairly represent all potential camera settings. For simplicity, we decide the focal length to be one. The principal point is fixed at zero, as the translation vector and the principal point serve similar functions. Using the camera parameters $c = R, T$, we define the projection function $\mathbf{P}$ that projects the self-created pose $\mathcal{P}^{3D}$ to a 2D pose $\mathcal{P}^{2D} \in \mathbb{R}^{J \times 2}$ following Drover *et al.* [26] as:

$$\mathbf{P}(c, \mathcal{P}^{3D}) = \mathcal{P}^{2D}. \tag{3}$$

The projected 2D pose, along with the original 3D pose, constitutes the final output of our generator. Since this pose is generated solely using biomechanical constraints, this approach ensures the reliability and diversity of the pose and produces data free from popularity bias.

## 3 Binary Depth Coordinates

Our goal is to mitigate the depth ambiguity that arises when lifting a 2D pose to 3D. Typical human pose estimators predict poses in the Camera Space Coordinates (CSC) system, which neglects perspective, often resulting in substantial projection errors. An alternative, the Normalized Device Coordinates (NDC), remaps the viewing frustum to a cube but still struggles with the inherent depth ambiguity in continuous space. The distinction between potential poses in continuous space is not merely a matter of measurement but one of understanding the multidimensional nature of human motion, where multiple poses can be correct in different contexts. For instance, a hand extended forwards towards the camera or backwards away from it could represent equally plausible positions in a given pose. However, the variance between such positions in continuous space is significant, introducing a level of complexity that can obfuscate the learning process, as the model must recognize and account for fundamentally different yet anatomically feasible poses that share similar 2D projections.

To address this, we introduce the Binary Depth Coordinates (BDC), a novel framework that accommodates the binary nature of depth in images and the continuity of real-world space. The BDC encodes a joint's position using a tuple $(x_i^{2D}, y_i^{2D}, l_i, s_i, P_i)$, where $x_i^{2D}$ and $x_i^{2D}$ specify the $i^{th}$ joint's location in the image plane. The additional elements $l_i$, representing the bone length, and $s_i$, a binary depth parameter indicating depth relative to the plane of the image, provide the necessary context for a mapping function to estimate the corresponding 3D coordinates $(x_i^{3D}, y_i^{3D}, z_i^{3D})$ within the CSC. The parameter $P_i = [P_i^x, P_i^y, P_i^z]$ stores the 3D coordinates of the preceding joint in the skeletal tree, helping to maintain the anatomical structure during transformations. In the BDC, the 3D depth $z_{3D}$ is computed by solving a quadratic equation derived from the geometric constraints of

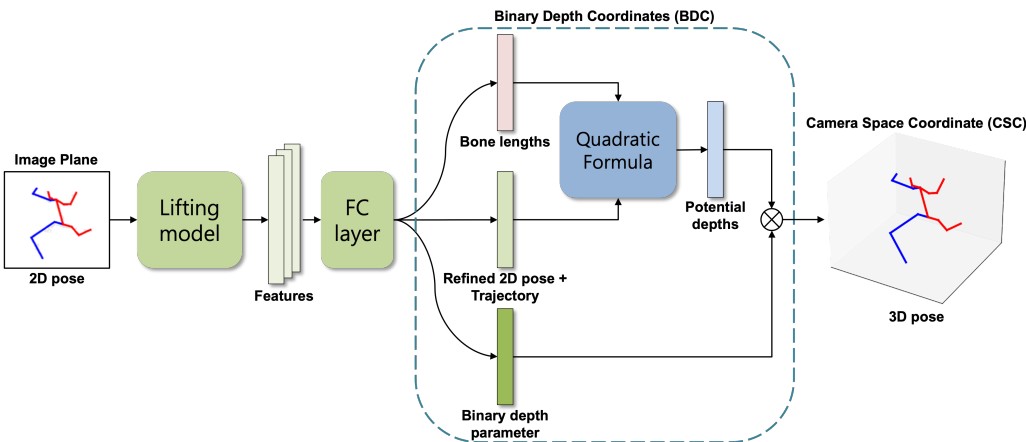

Figure 4: Our estimation process using BDC begins by using a fully connected layer to generate the bone lengths $l$, refined 2D joint, trajectoriy $z_0^{3D}$, and the binary depth parameters s from the input 2D joint positions. These bone lengths and 2D trajectories are then processed through the quadratic formula to determine possible depth values ($z^{3D+}$, $z^{3D-}$). The final 3D pose in the camera coordinate system is obtained by combining the binary depth parameter with the possible depth values, resulting in the accurate reconstruction of the 3D skeleton.

the joint's location in space. The equation is formulated as follows:

$$(x_i^{3D} - P_i^x)^2 + (y_i^{3D} - P_i^y)^2 + (z_i^{3D} - P_i^z)^2 = l_i^2, \qquad(4)$$

where $x_i^{3D}$ and $y_i^{3D}$ are the coordinates in the CSC calculated by back-projecting a ray from the image plane using the camera parameters:

$$x_i^{3D} = \frac{x_i^{2D} - c_x}{f_x} \cdot z_i^{3D}, y_i^{3D} = \frac{y_i^{2D} - c_y}{f_y} \cdot z_i^{3D}. \qquad(5)$$

Here, $c_x$ and $c_y$ denote the camera principal points, while $f_x$ and $f_y$ represent the focal lengths along the x and y axes, respectively. The solution to the quadratic equation provides two potential values for $z_i^{3D}$ as $z_i^{3D+}$ and $z_i^{3D-}$. These values reflect the dual nature of depth perception in perspective projection. To resolve the ambiguity between these two depths, the binary depth parameter $s_i$ is utilized. This parameter, derived from the BDC, indicates whether the joint is closer to or further from the camera relative to its connected joint. Specifically: When $s_i = 1$, it is interpreted that the joint is closer to the camera, thus selecting $z_i^{3D+}$ as the depth value. Conversely, when $s_i = -1$, the system chooses $z_i^{3D-}$, indicating that the joint is positioned further away from the camera. Our method precisely maps 2D poses to their accurate 3D configurations, leveraging observed and computed depth cues to resolve ambiguities and maintain anatomical consistency. The BDC integrates perspective projection through binary parameters and detailed bone length measurements, facilitating the accurate reconstruction of skeletal structure and overcoming depth ambiguity. By accurately calculating skeletal bone lengths and relationships, BDC enables the precise estimation of 3D poses, demonstrating a significant advancement over traditional coordinate systems that fall short in such complex scenarios.

## 3.1 Integration of BDC with Existing 3D HPE Models

The BDC is designed to integrate seamlessly into existing 3D human pose estimation frameworks with minimal modifications. This compatibility is achieved by adapting BDC to the final layers of commonly used neural network architectures. By adjusting the dimensionality of the output layer to accommodate our parameters, BDC enables the prediction of 2D poses, bone lengths, and the binary depth parameter $s_i$, thereby facilitating its integration into existing models. Additionally, predicting $s_i$ independently for each joint may fail to capture the complex interdependencies between different body parts adequately. To address this, we predict $s_i$ collectively for anatomically grouped segments—specifically, the left arm, right arm, right leg, left leg, and torso. For example, when

predicting $s_i$ for the left arm, which typically involves three joints, there are $2^3 = 8$ possible combinations of depth configurations. Thus, the model's task is to select one of these eight possible states. To achieve this, we utilize the Gumbel-Softmax, which allows differentiable sampling from discrete probability distributions. This approach enables the model to efficiently manage the prediction complexity, enhancing both the accuracy and anatomical consistency of the 3D pose reconstructions. Figure 4 illustrates the integration of the BDC into the existing lifting model.

### 3.2 Depth Ambiguity Ratio

This section introduces a Depth Ambiguity Ratio (DAR) to quantitatively express the uncertainty in depth for each joint in a 3D pose. The DAR quantifies the uncertainty by normalizing the sum of the potential depth values, $z_i^{3D+}$ and $z_i^{3D-}$, obtained from the quadratic equation solving for depth. To ensure the ratio falls between 0 and 1, we normalize this sum by dividing it by twice the bone length, denoted as $2l_i$:

$$\text{DAR} = \frac{1}{J-1} \sum_{i=2}^{J} \frac{z_i^{3D+} + z_i^{3D-}}{2l_i}. \tag{6}$$

This normalized metric serves as a universal index to measure depth uncertainty, enhancing the analysis of 3D modeling accuracy and consistency across different implementations.

## 4 Related Work

**3D human pose estimation.** Initial approaches [27, 28, 29, 30, 31, 32, 33, 34, 35, 36] in 3D human pose estimation attempted to extract 3D poses directly from monocular images without leveraging 2D pose estimations. However, with significant progress in 2D pose estimation techniques [37, 38, 39], the method of lifting 2D poses to 3D has emerged as a predominant strategy due to its effectiveness and scalability. Our work focuses on advancing these lifting-based frameworks [40, 24, 41, 42], particularly in terms of scalability and generalization to diverse real-world scenarios.

To enhance the lifting-based framework, several complementary approaches have been explored. Leveraging temporal information apporches [42, 24, 43] across multiple frames is a critical area in 3D HPE, as it can improve the accuracy and smoothness of 3D pose predictions. [24] utilized temporal convolutions over sequences of 2D keypoints to model temporal dependencies and reduce ambiguities inherent in single-frame predictions. Another effective strategy involves using rich image features. Approaches such as [41] incorporate spatial relationships between keypoints directly from image features, enhancing the pose estimation process. Addressing the inherent depth ambiguity in 3D pose estimation, multi-hypothesis approaches [12, 44, 45] generate multiple plausible 3D pose predictions and select the best fit. These methods use an evolutionary training data approach to generate multiple hypotheses for 3D poses from monocular images, thereby handling ambiguity and improving robustness.

**Cross-domain learning for 3D HPE.** Generalizing to rare or unseen poses presents a significant challenge. Various strategies have been developed to address this by adapting training methods. For example, [28] introduced a method for optimal network architecture selection for different body parts, while [46] enhanced robustness by processing local regions of the 2D pose independently. [47] proposed inference stage optimization to extract distributional knowledge specific to target scenarios. Despite these advancements, they often struggle with the inherent limitations of training datasets, and self-supervised training frequently falls short in performance. To further enhance the generalization ability of 3D HPE, extensive research has focused on pose augmentation methods. Given the limitations of existing pose datasets and the challenges in collecting 3D pose data, generating additional 2D-3D pose pairs has proven beneficial. [28] suggested using body-part crossover and mutation operations to create new 3D poses, though constrained by dependence on the source dataset and fixed mutation ranges. [48] employed reinforcement learning for joint-angle rotation control without original datasets, but this self-supervised approach's performance is limited. [15] and [13] used GANs for pose augmentation, showing improved cross-dataset performance, primarily due to variations in camera viewpoints and human positions rather than increased pose diversity. [49] generated 2D-3D pose pairs from 2D pose labels of the target domain, closely matching its distribution. [16] introduced network designs like interpolation sub-net and body-part grouping net. Despite these advances, generated datasets often lack diversity, remain heavily dependent on the

source dataset, and do not adequately consider the pose estimator's adaptability to the generated data, leading to potential training instability with complex poses.

Our work focuses on pose augmentation for 3D HPE. By synthesizing novel 3D poses and integrating them with source poses, it increases pose diversity and mitigates depth ambiguity issues.

# 5 Experiments

Our experiments were conducted on the Human3.6M (H36M) [20], MPI-INF-3DHP (3DHP) [21], and 3DPW [22] benchmarks. H36M [20], a widely used benchmark for 3D human pose estimation, includes over 3.6 million indoor video frames captured from four angles with 7 subjects denoted as S1, S5, S6, S7, S8, S9, and S11. 3DHP [21] consists of 1.3 million images and human pose data captured from 8 subjects using a markerless motion capture system in a green screen studio with 14 cameras, with six indoor and outdoor sequences evaluated following standard protocols. 3DPW [22] dataset is a challenging collection of in-the-wild 2D and 3D data, uniquely captured using a moving camera, unlike other datasets. We evaluate pose accuracy on H36M and 3DPW using mean per joint position error (MPJPE) and procrustes aligned mean per joint position error (PMPJPE), and on 3DHP with MPJPE, percentage of correct keypoints (PCK) within 150mm, and area under the curve (AUC), following established protocols.

## 5.1 Comparison on Pose Augmentation methods

In this section, we assess the generalizability of the data generated by our Biomechanical Pose Generator (BPG) in comparison to previous Pose Augmentation (PA) methods. Consistent with prior research and previous works [13, 16, 15], we utilize the single-frame VPose [24] network to lift 2D poses to 3D. Our training sources include subjects S1, S5, S6, S7, and S8 from the H36M dataset, and we employ a 16-keypoint human model for evaluation. The volume of synthesized poses matches that of the source data.

**Source-dataset evaluation results.** We conducted a series of experiments to assess the impact of integrating poses synthesized by the BPG with source data on the performance within the source dataset. For the evaluation, we utilized all available training data from the source dataset and employed HRnet [37] to generate predicted 2D keypoints as input, ensuring consistency with prior approaches. As depicted in Table 1, our method achieves improvements in MPJPE by 3.3mm and in P-MPJPE by 2.5mm. Furthermore, it is evident that using only the BPG still results in performance improvements over traditional methods. These results suggest that our synthesized data can produce a more diverse set of poses compared to traditional pose augmentation techniques. Furthermore, we examined the feasibility of achieving high accuracy with a reduced amount of training data. For this evaluation, we utilized subsets of the source dataset, specifically S1 or S1 + S5,

Table 1: Results on the H36M dataset, using the full set of training data. S9 and S11 are the targets.

| Method | PA | MPJPE↓ | P-MPJPE↓ |
|---|---|---|---|
| VPose [24] | | 52.7 | 40.6 |
| EvoSkeleton [50] | ✓ | 50.9 | 38.0 |
| PoseAug [13] | ✓ | 50.2 | 39.1 |
| DH-AUG [15] | ✓ | 49.8 | 38.3 |
| CEE-Net [16] | ✓ | 47.3 | 36.8 |
| BPG | ✓ | 46.9 | 36.5 |
| BPG + BDC | ✓ | **44.0** | **34.3** |

Table 2: Results on the H36M dataset, using a subset of the training data. We report MPJPE for evaluation.

| Method | PA | S1 | S1 + S5 |
|---|---|---|---|
| VPose [24] | | 65.2 | 57.9 |
| EvoSkeleton [50] | ✓ | 61.5 | 54.6 |
| PoseAug [13] | ✓ | 56.7 | 51.3 |
| DH-AUG [15] | ✓ | 52.2 | 47.0 |
| CEE-Net [16] | ✓ | 51.9 | 46.7 |
| BPG | ✓ | 51.5 | 46.1 |
| BPG + BDC | ✓ | **46.8** | **41.2** |

and used ground truth 2D keypoints as input, maintaining consistency with previous methods. As illustrated in Table 2, our experiments demonstrate that leveraging medical knowledge to generate poses allows our method to significantly enhance performance over existing methods, even with less training data. Notably, the relative performance gains observed with reduced training data were substantially greater than those achieved with the full training dataset. Furthermore, it is evident that using only the BPG still results in performance improvements over traditional methods. This finding underscores the robustness of our approach, which maintains high accuracy and significantly outperforms conventional methods even when the training data is limited.

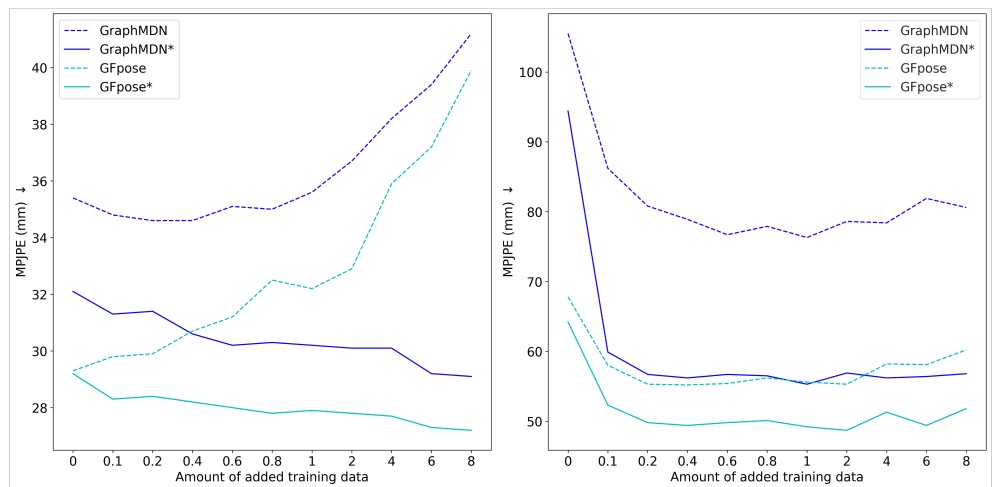

Figure 5: Results of added training data on MPJPE for GraphMDN [45] and GFpose [44]. *: Indicates that the model uses the BDC. Left: source dataset (H36M). Right: cross-domain (3DHP). The x-axis shows the ratio of added training data relative to the original dataset size (0 to 8).

**Cross-dataset evaluation results.** To evaluate whether the poses synthesized by the BPG contribute to improved generalization performance, we conducted experiments in a cross-dataset environment using the 3DPW and MPI-INF-3DHP datasets. For the evaluation, we utilized all available training data from the source dataset and employed ground truth 2D keypoints as input, consistent with previous methods. Our approach demonstrated performance improvements across both datasets. As shown in Table 3, our method improved PCK by 2%, AUC by 7%, and MPJPE by 7.9mm. Notably, significant performance gains were observed in the 3DPW dataset, which features outdoor scenes with moving cameras. As indicated in Table 4, our approach achieved improvements of 15.7mm in P-MPJPE. Furthermore, it is evident that using only the BPG still results in performance improvements over traditional methods. These results underscore the effectiveness of our synthesized data in enhancing generalization performance, particularly in challenging real-world environments.

Table 3: Cross-dataset evaluation results on 3DHP dataset.

| Method | PA | PCK↑ | AUC↑ | MPJPE↓ |
|---|---|---|---|---|
| VPose [24] | | 80.9 | 42.5 | 102.3 |
| EvoSkeleton [50] | ✓ | 81.2 | 46.1 | 99.7 |
| RepNet [51] | ✓ | 81.8 | 54.8 | 92.5 |
| PoseAug [13] | ✓ | 88.6 | 577.3 | 73.0 |
| DH-AUG [15] | ✓ | 89.3 | 57.9 | 71.2 |
| PoseGU [14] | ✓ | 86.3 | 57.2 | 75.0 |
| CEE-Net [16] | ✓ | 89.9 | 58.2 | 69.7 |
| BPG | ✓ | 91.2 | 58.3 | 69.1 |
| BPG + BDC | ✓ | **91.8** | **62.3** | **61.8** |

Table 4: Cross-dataset evaluation results on 3DPW dataset.

| Method | PA | P-MPJPE↓ | MPJPE↓ |
|---|---|---|---|
| VPose [24] | | 94.6 | 125.7 |
| VIBE [27] | ✓ | 81.6 | 122.5 |
| PoseAug [13] | ✓ | 81.6 | 119.0 |
| DH-AUG [15] | ✓ | 79.3 | 112.8 |
| PoseGU [14] | ✓ | 92.3 | - |
| CEE-Net [16] | ✓ | 76.8 | - |
| BPG | ✓ | 66.0 | 96.5 |
| BPG + BDC | ✓ | **61.1** | **90.3** |

## 5.2 Analysis on BPG and BDC.

We evaluate the performance of our BDC by comparing poses across varying levels of depth ambiguity, using the proposed depth ambiguity ratio to quantify and analyze the results. In this experiment, we utilize multi-hypothesis model [45] for 3D HPE, which are designed to address depth ambiguity. As illustrated in Figure 2, baseline model exhibit decreased performance on poses with high depth ambiguity. This trend is exacerbated when additional training data is introduced. However, when applying the BDC to existing models, performance remains consistent regardless of depth ambiguity. Furthermore, adding more data improves performance, even for poses with increased depth ambiguity. This observation highlights the robustness of the BDC in mitigating depth ambiguity problem and enhancing overall model accuracy.

Table 5: Ablation study with different BPG strategies on testing H36M, where the model was trained exclusively on generated data.

| Method | NROM | w/o RR | Ratio | PC | MPJPE | P-MPJPE |
|---|---|---|---|---|---|---|
| Kinematic model | | | | | 122.3 | 106.2 |
| Variant A | ✓ | | | | 65.7 | 56.6 |
| Variant B | ✓ | ✓ | | | 64.2 | 54.3 |
| Variant C | ✓ | | ✓ | | 62.6 | 53.9 |
| Variant D | ✓ | ✓ | ✓ | | 61.9 | 52.3 |
| BPG | ✓ | ✓ | ✓ | ✓ | **59.8** | **50.6** |

**Scalability of integrating BDC and BPG.** In this experiment, we analyze performance trends as the size of the training dataset increases using BPG. We utilize a multi-hypothesis model [45] for 3D HPE, designed to address depth ambiguity. By leveraging BPG, we expand the dataset size based on a fixed source dataset. As shown in Figure 5, increasing the dataset size using BPG enhances cross-domain performance. However, for baseline models, performance on the source dataset declines as the dataset size increases, indicating potential overfitting and increased pose complexity. Conversely, integrating BDC with baseline models results in improved cross-domain performance and consistent performance gains on the source dataset as the dataset size increases. This highlights the effectiveness of BDC in enhancing model robustness and scalability, mitigating the challenges of depth ambiguity, and improving overall performance in both source and cross-domain evaluations.

**BDC effects on 3D HPE models.** We investigated whether utilizing the BDC in state-of-the-art (SOTA) 3D HPE models across various tasks could lead to higher accuracy. For the training phase, we utilized all available training data from the source dataset and adhered to each model's specific training protocols to ensure consistency and fairness in evaluation.

Table 6: Results on H36M dataset. S9,S11 are the target.

| Task | Method | MPJPE | P-MPJPE |
|---|---|---|---|
| Multi-hypo | GraphMDN [45] | 46.2 | 36.3 |
| | GraphMDN* | 41.7 | 30.9 |
| | GFpose [44] | 36.6 | 30.5 |
| | GFpose* | 35.1 | 29.3 |
| Image features | Conpose [41] | 41.6 | 33.9 |
| | Conpose* | 38.5 | 33.8 |
| Multi-frame | MixSTE [42] | 40.9 | 32.6 |
| | MixSTE* | 38.3 | 32.2 |

As shown in Table 6, our approach consistently improved performance across all tasks in source dataset evaluations. The integration of BDC resulted in measurable improvements in key metrics, demonstrating the robustness and effectiveness of our method in enhancing model accuracy.

**Ablation study on BPG.** In our ablation study, we evaluated various strategies within our framework, including normal range of motion (NROM), rotational restriction (RR), body ratio variation, and Pose Confidence (PC). Only data synthesized with the BPG was used as training data. The results presented in Table 5 highlight a consistent trend in performance enhancement as each strategy is successively incorporated. Specifically, NROM, as indicated by Variant A, has the most significant influence on performance. However, both Variants A and C indicate that introducing RR can somewhat diminish performance. The findings from Variants C and D emphasize the benefits of incorporating diverse pose proportions for better generalization. Finally, the outcome of Variant D and BPG warn of the pitfalls of including physically improbable postures without PC, suggesting that such postures can hinder effective learning. From these observations, it is clear that the confluence of all strategies is essential for achieving peak performance.

**Ablation study on BDC.** In this ablation study, we compare the performance of various coordinate systems on a multi-hypothesis model [45] for 3D HPE, designed to address depth ambiguity. Specifically, we evaluate the Camera Space Coordinate (CSC), Normalized Device Coordinate (NDC), and BDC systems. The results, presented in Table 7, highlight the effectiveness

Table 7: Results on H36M dataset. S9,S11 are the target.

| Method | MPJPE | P-MPJPE |
|---|---|---|
| CSC | 46.2 | 36.3 |
| NDC | 44.3 | 32.4 |
| BDC w/o GS | 42.6 | 31.5 |
| BDC | 41.7 | 30.9 |

of the BDC system in improving model performance. The CSC and NDC systems serve as baselines for comparison. Additionally, we examine the impact of applying BDC without and with grouped segments (GS). As indicated, the BDC system, particularly when integrated with GS, demonstrates superior performance in reducing both MPJPE and P-MPJPE. This underscores the potential of BDC with GS in addressing depth ambiguity and enhancing the overall accuracy of 3D HPE models.

# 6 Conclusion

In this study, we introduced the BDC and the BPG to address key challenges in 3D HPE, particularly the issues of dataset diversity and depth ambiguity. Our approach leverages biomechanical principles and a binary classification framework to generate a wide array of plausible 3D poses and simplify depth estimation, respectively.

We demonstrated that integrating BDC and BPG into state-of-the-art 3D HPE models significantly improves accuracy and generalization across various datasets and tasks. Experiments on the H36M, 3DHP, and 3DPW datasets showed consistent performance enhancements, indicating that our synthesized data offers substantial benefits over traditional pose augmentation techniques, leading to more reliable 3D pose estimations in diverse scenarios.

**Limitations.** Despite the promising results, our approach has certain limitations. The BPG generates poses based on single-frame data only, missing temporal dynamics crucial for motion analysis. Extending BPG to handle multi-frame data could generate more realistic poses. Additionally, while integrating BDC and BP enhances model performance, generating additional data beyond a certain point does not lead to further improvements, suggesting a saturation point. Future work should optimize the amount and type of synthetic data to maximize performance without unnecessary computational overhead.

## Acknowledgements

This work was supported by Institute of Information & communications Technology Planning & Evaluation (IITP) grant funded by the Korea government(MSIT) (No. RS-2019-II190079, Artificial Intelligence Graduate School Program (Korea University), No. 2021-0-02068, Artificial Intelligence Innovation Hub, No. RS-2024-00457882, AI Research Hub Project, and No. RS-2024-00336673, AI Technology for Interactive Communication of Language Impaired Individuals).

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

# Appendix

## A Implementation Details

 For the comparison of pose augmentation methods, without BDC, we utilize the single-frame version of VPose [24] as the pose estimator. When applying BDC, we introduce four 1D convolution layers at the end of VPose [24] to estimate parameters such as root depth, 2D pose, bone length, and binary depth. In the case of the multi-hypothesis model [45], we create four fully connected layers to estimate the same parameters: root depth, 2D pose, bone length, and binary depth. The training loss remains consistent with the losses used in the original models. Specifically, VPose [24] employs Mean Squared Error Loss (MSELoss), while the multi-hypothesis model [45] uses Mixture Density Network (MDN) loss.

We train the models on the H36M dataset for 50 epochs with a batch size of 1024. All components are optimized using the Adam optimizer with an initial learning rate of 0.001, which linearly decays over time. For the BPG, training is conducted on a single Nvidia GTX 3090 Ti GPU with a batch size of 1024 for 100 epochs, taking approximately two days to complete the experiments. The Adam optimizer is utilized with an initial learning rate of 3e-4, which decreases linearly during the training process. To stabilize the training process and enhance performance, we pre-train the models on synthetic data before end-to-end training. This pre-training phase ensures that the models are better initialized and capable of more robust learning during the comprehensive training phase.

## B Visualization

We conducted extensive experiments to evaluate the impact of our synthesized data using BPG on 3D HPE models. Figure 6 shows that our synthesized poses occupy a significantly larger area compared to the existing dataset. However, this did not immediately translate into performance improvement; rather, it resulted in a performance decline. To identify the cause of this decline, we examined the per-frame performance of the baseline model's poses in Figure 5.A. The results indicated that performance degradation occurred in most scenarios. Nonetheless, there was a promising aspect: in environments where the baseline model failed, our approach showed performance improvement. Furthermore, when integrating the BDC into the baseline model, we observed performance enhancement across almost all regions. In Figure 5.B, the visualization of the areas where performance declined after pose augmentation revealed that the results of complex poses appeared with flattened depth. This indicates that the increased pose diversity led to heightened depth ambiguity, causing predictions to average towards the center. However, by utilizing our BDC, we observed robust predictions even with increased pose diversity, indirectly mitigating depth ambiguity.

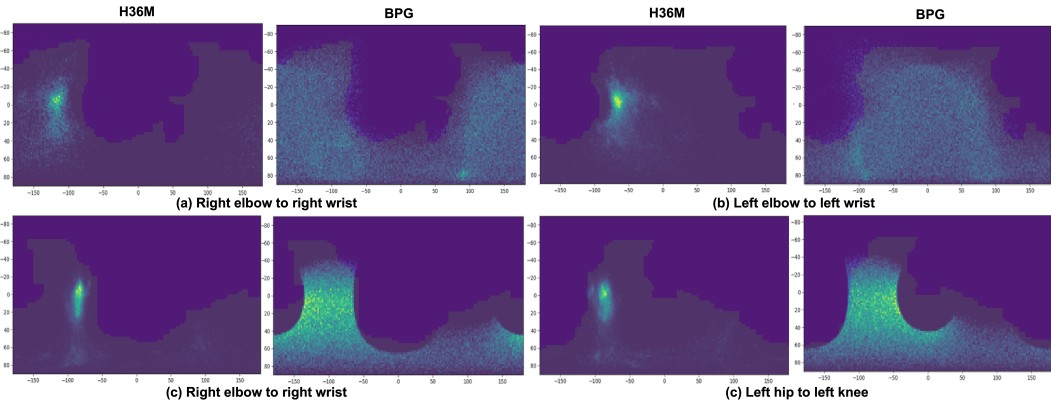

Figure 6: Spherical coordinate distribution for the vector across datasets: H36M [20] and BPG, with valid regions highlighted in grey (taken from [23]).

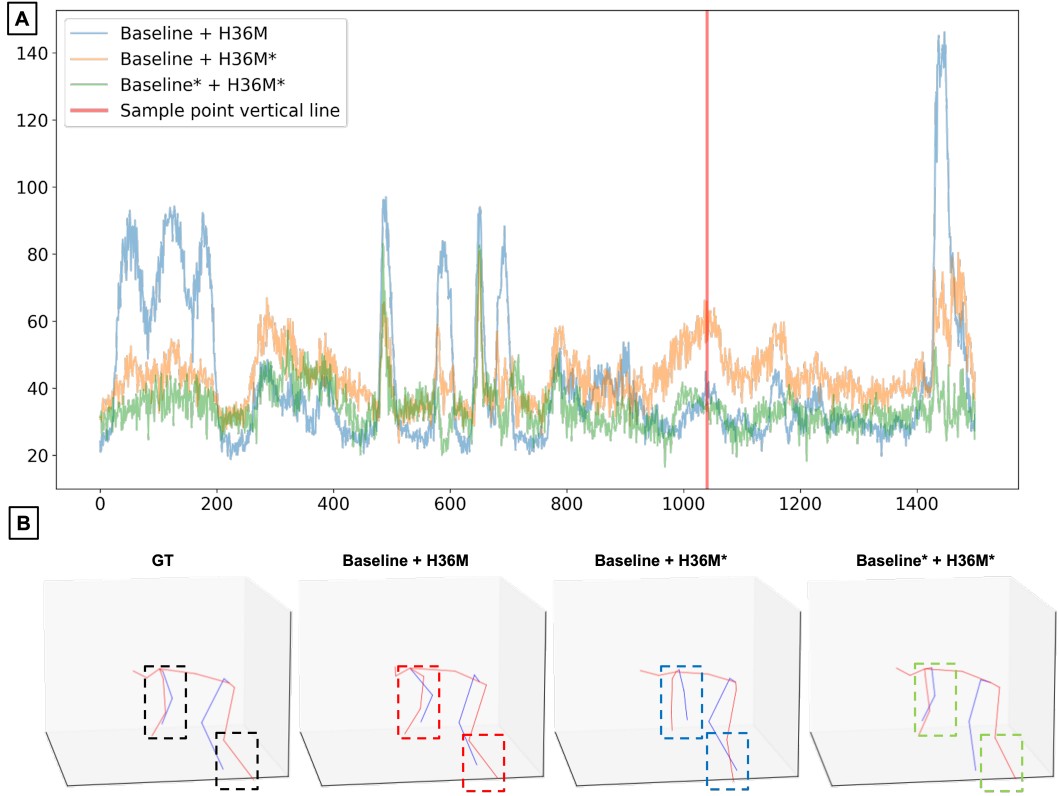

Figure 7: A : Frame-wise comparison on H36M. Baseline* employs BDC on the Baseline model [45], and H36M* combines source data with additional data synthesized by BPG, expanding the total dataset to ten times the H36M training size. B: The result of visualizing the poses at the vertical line in Figure 7.A.

## C    Formula details

### C.1    Forward Kinematics Function.

$\mathbf{T}(\mathbf{N}_{pose} = \{l, \theta\})$ takes a joint variable vector $\boldsymbol{\theta} = [\theta_1, \theta_2, \ldots, \theta_n]$ and a link length vector $\mathbf{l} = [l_1, l_2, \ldots, l_n]$ as input, and returns a transformation matrix representing the position and attitude of the end effector. The mathematical representation of this is

$$\mathbf{T}(\boldsymbol{\theta}, \mathbf{l}) = \begin{bmatrix} \mathbf{R}(\boldsymbol{\theta}, \mathbf{l}) & \mathbf{d}(\boldsymbol{\theta}, \mathbf{l}) \\ \mathbf{0}^\top & 1 \end{bmatrix}$$

where $\mathbf{R}(\boldsymbol{\theta}, \mathbf{l})$ is a $3 \times 3$ rotation matrix, and $\mathbf{d}(\boldsymbol{\theta}, \mathbf{l})$ is a $3 \times 1$ position vector. This matrix represents the position and attitude of the end effector in the robot's reference frame.

The forward kinematics function is defined as follows

$$\mathbf{T}(\boldsymbol{\theta}, \mathbf{l}) = \mathbf{A}_1(\theta_1, l_1)\mathbf{A}_2(\theta_2, l_2) \cdots \mathbf{A}_n(\theta_n, l_n)$$

where $\mathbf{A}_i(\theta_i, l_i)$ is the individual transformation matrix for each joint $i$. This transformation matrix represents the rotation and displacement of each joint, and typically has the form

$$\mathbf{A}_i(\theta_i, l_i) = \begin{bmatrix} \cos(\theta_i) & -\sin(\theta_i) & 0 & l_i \\ \sin(\theta_i)\cos(\alpha_i) & \cos(\theta_i)\cos(\alpha_i) & -\sin(\alpha_i) & -d_i\sin(\alpha_i) \\ \sin(\theta_i)\sin(\alpha_i) & \cos(\theta_i)\sin(\alpha_i) & \cos(\alpha_i) & d_i\cos(\alpha_i) \\ 0 & 0 & 0 & 1 \end{bmatrix}$$

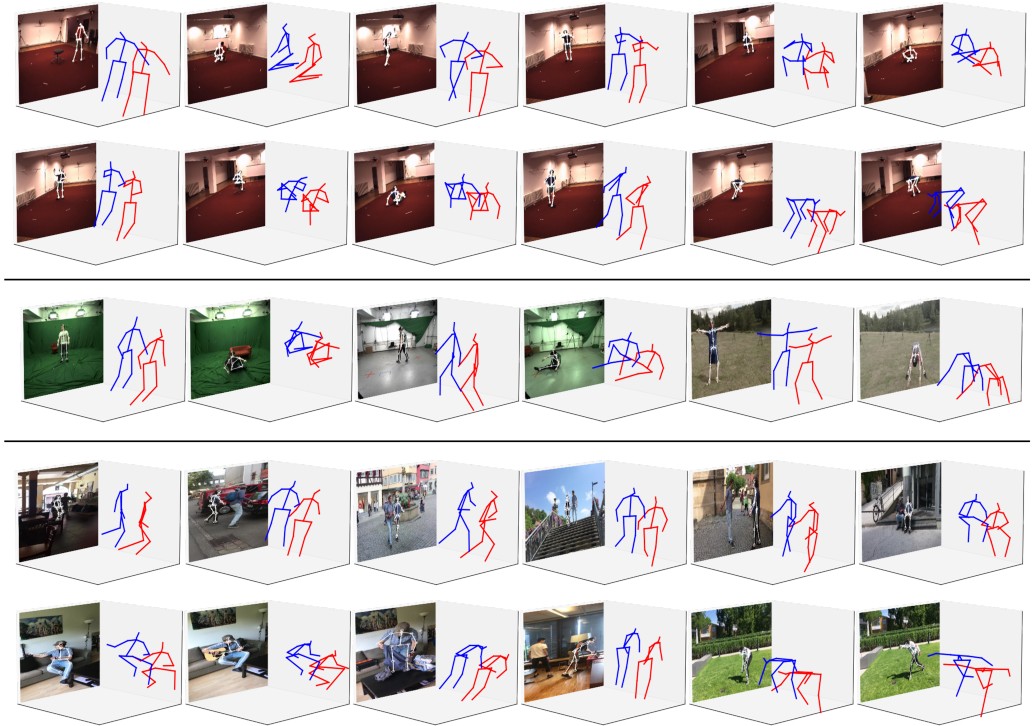

Figure 8: Qualitative results of ground truth poses (blue) and predicted poses (red) on various datasets, H36M (rows 1, 2), 3DHP (row 3), and 3DPW (rows 4, 5).

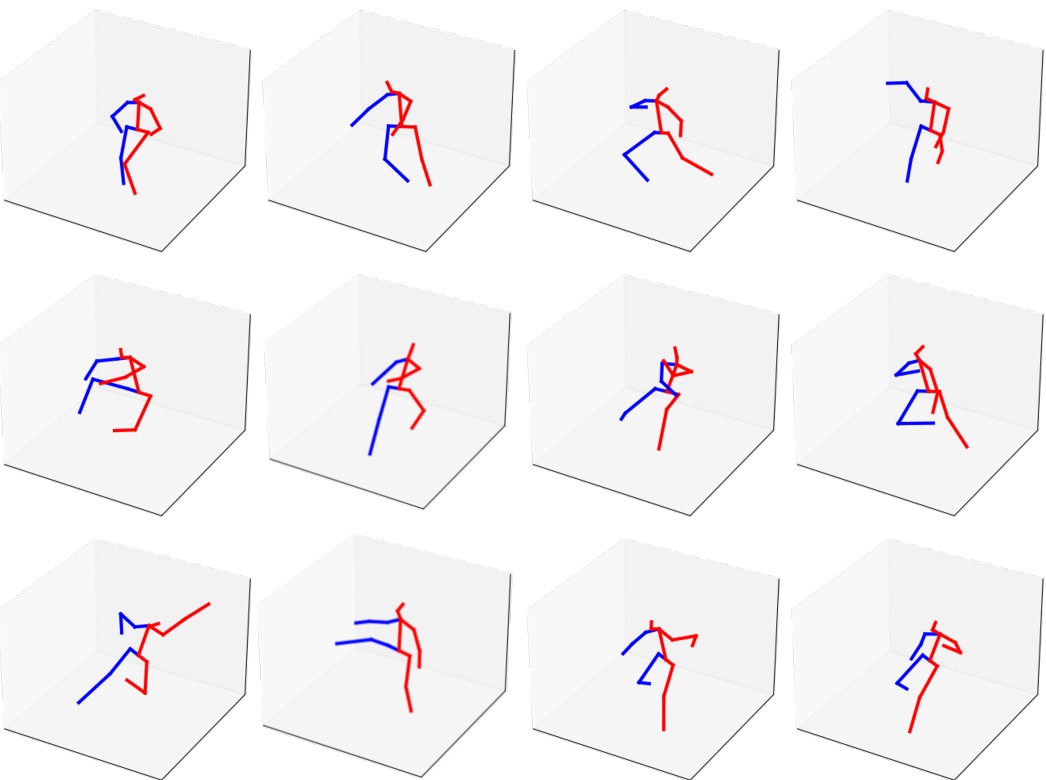

Figure 9: Visualization of synthesized poses with our BPG.

where $l_i$ is the length of link $i$, and $d_i$ and $\alpha_i$ are the Denavit-Hartenberg (DH) parameters of the robot.

So, the full formula for the forward kinematics function is

$$\mathbf{T}(\boldsymbol{\theta}, \mathbf{l}) = \prod_{i=1}^{n} \mathbf{A}_i(\theta_i, l_i)$$

This formula represents sequentially multiplying the individual transformation matrices for each joint to compute the overall transformation matrix. This allows us to get the position and pose of the end effector.

The forward kinematics function $\mathbf{T}(\boldsymbol{\theta}, \mathbf{l})$ takes a joint variable vector $\boldsymbol{\theta} = [\theta_1, \theta_2, \ldots, \theta_n]$ and a link length vector $\mathbf{l} = [l_1, l_2, \ldots, l_n]$ as input, and returns a transformation matrix representing the position and attitude of the end effector. The mathematical representation of this is

$$\mathbf{T}(\boldsymbol{\theta}, \mathbf{l}) = \begin{bmatrix} \mathbf{R}(\boldsymbol{\theta}, \mathbf{l}) & \mathbf{d}(\boldsymbol{\theta}, \mathbf{l}) \\ \mathbf{0}^\top & 1 \end{bmatrix}$$

where $\mathbf{R}(\boldsymbol{\theta}, \mathbf{l})$ is a $3 \times 3$ rotation matrix, and $\mathbf{d}(\boldsymbol{\theta}, \mathbf{l})$ is a $3 \times 1$ position vector. This matrix represents the position and attitude of the end effector in the robot's reference frame.

The forward kinematics function is defined as follows

$$\mathbf{T}(\boldsymbol{\theta}, \mathbf{l}) = \mathbf{A}_1(\theta_1, l_1) \mathbf{A}_2(\theta_2, l_2) \cdots \mathbf{A}_n(\theta_n, l_n)$$

where $\mathbf{A}_i(\theta_i, l_i)$ is the individual transformation matrix for each joint $i$. This transformation matrix represents the rotation and displacement of each joint, and typically has the form

$$\mathbf{A}_i(\theta_i, l_i) = \begin{bmatrix} \cos(\theta_i) & -\sin(\theta_i) & 0 & l_i \\ \sin(\theta_i)\cos(\alpha_i) & \cos(\theta_i)\cos(\alpha_i) & -\sin(\alpha_i) & -d_i \sin(\alpha_i) \\ \sin(\theta_i)\sin(\alpha_i) & \cos(\theta_i)\sin(\alpha_i) & \cos(\alpha_i) & d_i \cos(\alpha_i) \\ 0 & 0 & 0 & 1 \end{bmatrix}$$

where $l_i$ is the length of link $i$, and $d_i$ and $\alpha_i$ are the Denavit-Hartenberg (DH) parameters of the robot.

So, the full formula for the forward kinematics function is

$$\mathbf{T}(\boldsymbol{\theta}, \mathbf{l}) = \prod_{i=1}^{n} \mathbf{A}_i(\theta_i, l_i) = P_{3\mathrm{D}}$$

This formula represents sequentially multiplying the individual transformation matrices for each joint to compute the overall transformation matrix. This allows us to get the position and pose of the end effector.

In summary, the forward kinematics function $\mathbf{T}$ maps the initial joint positions and link lengths (which constitute the kinematic model) to the 3D position and orientation of the end effector:

## C.2 The Indicator Function

Let $X$ be a set, and let $A$ be a subset of $X$. The indicator function $\mathbb{1}_A(x)$ of a subset $A$ of a set $X$ is defined as:

$$\mathbb{1}_A(x) = \begin{cases} 1 & \text{if } x \in A \\ 0 & \text{if } x \notin A \end{cases}$$

### C.3 Camera Projection Function.

The camera projection function $\mathbf{P}$ maps a 3D point $\mathbf{X} = [X, Y, Z, 1]^\top$ in homogeneous coordinates to a 2D point $\mathbf{x} = [u, v, 1]^\top$ on the image plane. The mathematical representation of this function is

$$\mathbf{x} = \mathbf{P}\mathbf{X}$$

where $\mathbf{P}$ is the camera projection matrix, defined as

$$\mathbf{P} = \mathbf{K}[R|T]$$

Here, $\mathbf{K}$ is the camera intrinsic matrix, $R$ is the rotation matrix representing the camera orientation, and $T$ is the translation vector representing the camera position. The intrinsic matrix $\mathbf{K}$ is given by

$$\mathbf{K} = \begin{bmatrix} f_x & 0 & c_x \\ 0 & f_y & c_y \\ 0 & 0 & 1 \end{bmatrix}$$

where $f_x$ and $f_y$ are the focal lengths in the x and y directions, respectively, and $c_x$ and $c_y$ are the coordinates of the principal point.

The full camera projection function is thus represented as

$$\mathbf{x} = \mathbf{K}[\mathbf{R}|\mathbf{t}]\mathbf{X}$$

This formula maps a 3D point $\mathbf{X}$ in the world coordinate system to a 2D point $\mathbf{x}$ on the image plane, taking into account the camera's intrinsic parameters and its pose in the world.

### C.4 Solving the quadratic equation for $z_i^{3D}$.

Given the equation:
$$(x_i^{3D} - P_i^x)^2 + (y_i^{3D} - P_i^y)^2 + (z_i^{3D} - P_i^z)^2 = l_i^2$$

and the transformations:
$$x_i^{3D} = \frac{x_i^{2D} - c_x}{f_x} \cdot z_i^{3D}, \quad y_i^{3D} = \frac{y_i^{2D} - c_y}{f_y} \cdot z_i^{3D}$$

we substitute these into the original equation:

$$\left( \frac{x_i^{2D} - c_x}{f_x} \cdot z_i^{3D} - P_i^x \right)^2 + \left( \frac{y_i^{2D} - c_y}{f_y} \cdot z_i^{3D} - P_i^y \right)^2 + (z_i^{3D} - P_i^z)^2 = l_i^2$$

Expanding and rearranging terms, we get a quadratic equation in $z_i^{3D}$:
$$A(z_i^{3D})^2 + Bz_i^{3D} + C = 0$$

where:
$$A = \frac{(x_i^{2D} - c_x)^2}{f_x^2} + \frac{(y_i^{2D} - c_y)^2}{f_y^2} + 1$$

$$B = -2 \left( \frac{(x_i^{2D} - c_x)P_i^x}{f_x} + \frac{(y_i^{2D} - c_y)P_i^y}{f_y} + P_i^z \right)$$

$$C = (P_i^x)^2 + (P_i^y)^2 + (P_i^z)^2 - l_i^2$$

The solutions for $z_i^{3D}$ are given by the quadratic formula:
$$z_i^{3D} = \frac{-B \pm \sqrt{B^2 - 4AC}}{2A}$$

Therefore, the two roots are:

$$z_i^{3D+} = \frac{-B + \sqrt{B^2 - 4AC}}{2A}$$

$$z_i^{3D-} = \frac{-B - \sqrt{B^2 - 4AC}}{2A}$$

# D  Pseudo-Code

---
**Algorithm 1** Biomechanical Pose Generator (BPG)
---
1: **Input:** Normal Range of Motion (NROM) intervals $\theta_{\text{lower}}$ and $\theta_{\text{upper}}$, mean limb length ratios $\bar{l}_j$, proportion range parameter $\alpha$, Camera intervals $T^{\text{lower}}$ and $T^{\text{upper}}$
2: **Output:** Generated 3D pose $P_{\text{3D}}$, projected 2D pose $P_{\text{2D}}$
3: Initialize pose parameters $N_{\text{pose}} \leftarrow \{\}$
4: **for** each bone $j$ **do**
5:   Sample $l_j \sim \mathcal{U}((1-\alpha)\bar{l}_j, (1+\alpha)\bar{l}_j)$
6:   $N_{\text{pose}}.\text{append}(l_j)$
7: **end for**
8: **for** each joint $i$ **do**
9:   Sample $\theta_i \sim \mathcal{U}(\theta_{\text{lower},i}, \theta_{\text{upper},i})$
10:   $N_{\text{pose}}.\text{append}(\theta_i)$
11: **end for**
12: $P_{\text{3D}} \leftarrow \mathbf{T}(N_{\text{pose}})$
13: **for** each bone pair $(n, m)$ **do**
14:   Calculate $d_{n,m}$
15:   **if** $d_{n,m} < \bar{d}_{n,m}$ **then**
16:     Reject pose $P_{\text{3D}}$ and resample
17:   **end if**
18: **end for**
19: Sample rotation matrix $R \sim \mathcal{U}(SO(3))$
20: Sample translation vector $T \in \mathbb{R}^3 \sim \mathcal{U}(T^{\text{lower}}, T^{\text{upper}})$
21: Set focal length $f = 1$
22: Set principal point $p_p = (0, 0)$
23: Define camera parameters $c = \{R, T, f, p_p\}$
24: Project $P_{\text{3D}}$ to 2D pose $P_{\text{2D}}$ using the projection function $p(c, P_{\text{3D}})$
25: **return** $P_{\text{2D}}, P_{\text{3D}}$
---

**Algorithm 2** Binary Depth Coordinates (BDC)

---

1: **Input:** 2D joint positions $\mathbf{P}_{2D}$, bone lengths $l$, binary depth parameter $s$, camera parameters $(c_x, c_y, f_x, f_y)$, root depth $z_0^{3D}$, Skeleton Tree $T$
2: **Output:** 3D joint positions $\mathbf{P}_{3D}$
3: Initialize $\mathbf{P}_{3D} \leftarrow \{\}$
4: Compute the 3D coordinates of the root joint:
5: $x_0^{3D} \leftarrow \frac{x_0^{2D} - c_x}{f_x} \cdot z_0^{3D}$
6: $y_0^{3D} \leftarrow \frac{y_0^{2D} - c_y}{f_y} \cdot z_0^{3D}$
7: Append $(x_0^{3D}, y_0^{3D}, z_0^{3D})$ to $\mathbf{P}_{3D}$
8: **for** each joint $i$ in tree order $T$ **do**
9:     Compute the 3D coordinates $x_i^{3D}, y_i^{3D}$ using:

$$x_i^{3D} = \frac{x_i^{2D} - c_x}{f_x} \cdot z_i^{3D} \tag{7}$$

$$y_i^{3D} = \frac{y_i^{2D} - c_y}{f_y} \cdot z_i^{3D} \tag{8}$$

10:     Solve the quadratic equation for $z_i^{3D}$:

$$(x_i^{3D} - P_{x,\text{parent}(i)})^2 + (y_i^{3D} - P_{y,\text{parent}(i)})^2 + (z_i^{3D} - P_{z,\text{parent}(i)})^2 = l_i^2 \tag{9}$$

11:     This yields two potential values: $z_i^{3D+}$ and $z_i^{3D-}$
12:     Use the binary depth parameter $s_i$ to select the correct depth:

$$z_i^{3D} = \begin{cases} z_i^{3D+} & \text{if } s_i = 1 \\ z_i^{3D-} & \text{if } s_i = -1 \end{cases} \tag{10}$$

13:     Compute $x_i^{3D}, y_i^{3D}$ with the selected $z_i^{3D}$:

$$x_i^{3D} = \frac{x_i^{2D} - c_x}{f_x} \cdot z_i^{3D} \tag{11}$$

$$y_i^{3D} = \frac{y_i^{2D} - c_y}{f_y} \cdot z_i^{3D} \tag{12}$$

14:     Append $(x_i^{3D}, y_i^{3D}, z_i^{3D})$ to $\mathbf{P}_{3D}$
15: **end for**
16: **return** $\mathbf{P}_{3D}$

---

