# OpenReview forum: "Toward Approaches to Scalability in 3D Human Pose Estimation"
_NeurIPS.cc/2024/Conference — NeurIPS 2024 poster_

### Official Review · Reviewer_Gv4b · 2024-07-11

**Soundness:** 3
**Presentation:** 2
**Contribution:** 3
**Rating:** 6
**Confidence:** 4

**Summary:**

Existing data in 3D human pose estimation are typically collected indoors with human actors. To address this scalability issue, the authors propose to synthesize 3D human pose data via an Osteo-kinematic model and introduce biochemical constraints for better physical plausibility. Additionally, to deal with the inherent ambiguity in single-view depth estimation, the authors introduce Binary Depth Coordinates to explicitly model the relative spatial relation between adjacent joints. Extensive experiments verify the effectiveness of the proposed approach.

**Strengths:**

1. Leveraging biomechanical prior knowledge to synthesize physically plausible human data is $\textbf{intuitive}$ and $\textbf{interesting}$.
2. Comprehensive experiments verify the effectiveness of the proposed data augmentation approach (BPG) and Binary Depth Coordinates (BDC). Specifically, BDC can be applied to different methods, e.g., image-based and lifting-based, showing superior generalization ability.

**Weaknesses:**

1. $\textbf{Repeated text}$: The first paragraph of Sec.2 appears to be a copy-paste from the abstract, which is highly discouraged.
2. $\textbf{Requirement of camera intrinsics}$: While BDC shows notable performance gains to baselines, solving depth requires camera intrinsics (principal point and focal length), typically not required by current 3D HPE methods. This requirement may introduce additional constraints for in-the-wild inference.

**Questions:**

1. In Fig. 2 and Fig.4, adding synthesized data consistently decreased performance for some baseline methods, e.g., GFpose. This seems counterintuitive to me. As the authors mentioned, overfitting might be a reason; do the authors have any other insights regarding this? Does this phenomenon indicate there is still a gap between the real data and synthesized data?

**Limitations:**

The authors have included the limitations in Sec.6. However, the requirement for camera intrinsics may also be considered a limitation and discussed.

---

> ### Author Rebuttal · Authors · 2024-08-05
>
> Thank you for your valuable and detailed review. Your feedback on the use of biomechanical knowledge and the practical limitations of our approach provides important guidance for further improving our work.
>
> ## Repetition in the First Paragraph of Section 2
>
> We sincerely thank the reviewer for pointing out the repetition in the first paragraph of Section 2. We appreciate your careful review and constructive feedback. We will ensure that this repetition is addressed and revised appropriately to enhance the clarity and originality of our manuscript. Thank you once again for your valuable input.
>
> ## Requirement of camera intrinsics
>
> We appreciate the reviewer's insightful comment regarding the requirement of camera intrinsics for solving depth with BDC. We understand that this may seem like an additional constraint compared to some current 3D HPE methods.
>
> However, we would like to clarify that the use of camera intrinsics in our equations primarily serves to illustrate the theoretical framework. In practice, similar to other research [1,2,3], we employ approximations for both inference and training, which effectively mitigate the need for precise camera intrinsic parameters. This approach allows our method to be applied more flexibly in in-the-wild scenarios without introducing significant constraints.
>
> [1] J. Sosa et al., "Self-supervised 3D Human Pose Estimation from a Single Image", CVPR 2023
>
> [2] B. Wandt et al., "ElePose: Unsupervised 3D Human Pose Estimation by Predicting Camera Elevation and Learning Normalizing Flows on 2D Poses", CVPR 2022
>
> [3] Z. Yu et al., "Towards alleviating the modeling ambiguity of unsupervised monocular 3d human pose estimation", ICCV 2021
>
> ## Insights on Decreased Performance with Synthesized Data
>
> We appreciate the reviewer's question regarding the decreased performance when adding synthesized data for some baseline methods, such as Gfpose, as shown in Figures 2 and 4.
>
> By examining Figure 6 (Appendix), we can provide additional insights into this phenomenon. It appears that the inclusion of excessive synthesized data can exacerbate the depth ambiguity problem for some baseline methods. As a result, these models tend to predict the middle ground rather than accurately differentiating between the front and back, leading to a degradation in performance.
>
> However, our Binary Depth Coordinates (BDC) approach mitigates this issue effectively. BDC's novel decomposition strategy and its handling of depth ambiguity ensure that the model maintains high performance even with the addition of synthesized data. This demonstrates the robustness of our method in addressing depth ambiguity and improving overall accuracy, unlike the observed degradation in baseline methods.
>
> ## **Concluding Remarks**
>
> We deeply appreciate the insightful comments provided by reviewer Gv4b and will thoughtfully incorporate their valuable suggestions in our rebuttal. We are confident that these responses will address the concerns raised and contribute to an improved evaluation of our manuscript.

---

> > ### Comment · Reviewer_Gv4b · 2024-08-10
> >
> > I would like to thank the authors' efforts to prepare the response. Most of my concerns were addressed.

---

### Official Review · Reviewer_696C · 2024-07-11

**Soundness:** 3
**Presentation:** 2
**Contribution:** 3
**Rating:** 5
**Confidence:** 4

**Summary:**

This paper introduces two components aimed at addressing challenges in 3D human pose estimation, specifically in terms of scalability and generalization. The authors propose a Biomechanical Pose Generator (BPG), which incorporates biomechanical principles to generate plausible new 3D poses. They also introduce Binary Depth Coordinates (BDC), a component designed to mitigate the depth ambiguity encountered when lifting a 2D pose to 3D. The paper includes ablation studies to demonstrate the impact of each component, and compares these new approaches to existing pose augmentation methods.

**Strengths:**

The paper’s focus on addressing the challenge of limited datasets and enhancing the generalizability of the method is interesting and to the best of my knowledge the idea of biomechanical pose generator which does not rely on a source dataset is novel. Also, the authors’ attention to the depth ambiguity in 3D pose estimation from a single image adds a value to the field. The authors have conducted comprehensive experiments and ablation studies, which provide valuable insights into the effectiveness of the proposed components. The inclusion of cross-dataset evaluation is crucial, as it allows for a robust assessment of the Biomechanical Pose Generator (BPG) component’s effectiveness.

**Weaknesses:**

1- The paper is generally well-written, but some parts could be clearer. Including a figure to illustrate the entire system could significantly help reader comprehension. For example, a diagram showing the VPose (or any baseline) architecture and the integration of the BDC component might be more effective than a text-only description. Additionally, including some implementation details about the BDC component in the main paper could improve the flow of information.

2- There are some ambiguities in the experiment section that need clarification. When referring to the “source-dataset”, it would be helpful to specify whether this refers to the Human 3.6M dataset or the newly synthesized poses. Similarly, when discussing evaluations on 3DHP and 3DPW, it would be beneficial to mention the specific subset used, such as the test set.

3- There appears to be some confusion between Table 1 and the results in Figure 4 (left). While Table 1 shows improvements in the Human 3.6M results when adding new poses generated from BPG, Figure 4 (left) indicates that adding more data increases the MPJPE error (without integrating BDC). This seems contradictory and could benefit from further explanation.

4- Typos: There are a couple of typographical errors that need correction. On Line 177, (xi) is repeated twice instead of yi. On Line 287, BDC should be corrected to BPG.

**Questions:**

As I mentioned in the weaknesses section, I would like to learn more about the effect of adding more synthesized poses to Human 3.6M and evaluating on the same source of data as currently the results in the Table 1 and Figure 4 are a bit confusing to me.

**Limitations:**

Yes

---

> ### Author Rebuttal · Authors · 2024-08-05
>
> Thank you for your comprehensive and constructive review. Your insights on the scalability and generalization aspects, along with suggestions for improved clarity, are invaluable and will be instrumental in enhancing our paper.
>
> ## Suggestions for Improving Clarity
>
> We appreciate the reviewer's positive feedback on the overall quality of our paper and their constructive suggestions for improving clarity. We agree that including a figure to illustrate the entire system could significantly aid reader comprehension. The diagram showing the baseline architecture and the integration of the BDC component has been added to the PDF linked in the General Response. Additionally, we have provided more detailed implementation specifics of the BDC component in the main paper. To aid understanding of the BDC, pseudo-code is included and can be found in Appendix D, Algorithm 2. These enhancements will make our manuscript clearer and more informative.
>
> ## Ambiguities in the Experiment Section
>
> We appreciate the reviewer's observation regarding ambiguities in the experiment section. We will carefully address these ambiguities and provide the necessary clarifications to improve the clarity and comprehensibility of our manuscript. Bringing this to our attention helps ensure that our revisions will enhance the overall quality of our work.
>
> ## Confusion between Table 1 and Figure 4 (left)?
>
> This is a valid point. We clarified this issue in the general response under "Confusion on Experimental Results.”
>
> ## Typos
>
> We acknowledge the reviewer's attention to detail regarding the typographical errors in our manuscript. We will correct the repeated (xi) on Line 177 to yi and change BDC to BPG on Line 287. Highlighting these issues helps us improve the accuracy and readability of our manuscript.
>
> ## **Concluding Remarks**
>
> We are grateful for the constructive feedback from reviewer 696C and will incorporate their valuable suggestions to refine our manuscript. We believe that our comprehensive responses will satisfactorily address the concerns and result in a favorable reassessment.

---

> > ### Comment · Reviewer_696C · 2024-08-10
> >
> > Thanks for the clarifications. I still believe that while the idea of the paper is novel in some aspects, the paper contains some ambiguities in several parts and it needs revision to be ready for a good publication. Given that and also according to other concerns raised by other reviewers, I'd rather keep my original rating.

---

### Official Review · Reviewer_9ZcB · 2024-07-11

**Soundness:** 3
**Presentation:** 2
**Contribution:** 2
**Rating:** 5
**Confidence:** 4

**Summary:**

The authors propose a 3D human pose estimation framework that incorporates data augmentation and depth ordering information. The main contributions are two-fold: First, the proposed Biomechanical Pose Generator (BPG) generates plausible body poses based on kinematic constraints, which is used for data augmentation. Second, the Binary Depth Coordinates (BDC) disambiguate the projective depth of each joint by classifying whether the joints are positioned towards or away from the camera. The proposed framework achieved state-of-the-art performance in single-frame 3D human pose estimation settings.

**Strengths:**

- The proposed method achieves state-of-the-art results in various 3D HPE datasets.
- The effect of data augmentation is validated in cross-domain learning settings.

**Weaknesses:**

My major concern lies on the novelty of the contribution.

- There are numerous research papers that regularize 3D human pose based on kinematic constraints. The authors did not clarify the distinctiveness of BPG from these conventional works, except for stating that BPG achieved better performance. An analysis showing how the proposed BPG generates more plausible poses compared to previous augmentation methods is required, either by displaying the generated poses or by showing qualitative estimation results.
- The concept of BDC is similar to [1] which learns ordinal depth information. The authors should cite the paper and discuss the difference.

The paper also contains	ambiguously explained parts or lacks details about their methods. Please refer to Questions section.

[1] G. Pavlakos et al., "Ordinal depth supervision for 3d human pose estimation", CVPR 2018

**Questions:**

Method
- In line 155, the focal length of the camera matrix is set to 1 for BPG, is it also the case for the datasets used? Or the camera matrix provided in the datasets are used?
- In line 179, what is the meaning of "depth relative to the plane of the image". I guess $s_i$ is the depth relative to the preceding joint not the image plane.

Experiments
- Why did the authors use different baseline architectures in Sec. 5.1 and 5.2?
- How much portion of augmented data from BPG used for experiments in Sec. 5.1?
- Given that using only BPG increases the error in Fig. 4 left, how could it be possible to achieve better performance in Table 1 and 2 when only BPG is used?
- Why didn't the authors use BPG in Table 6?
- What is the difference between Variant E and BPG in Table 8?

**Limitations:**

Suggestions
- In Fig. 4, it should be clearly stated what * means. It would be better to use GFpose+BDC instead of GFpose*.
- More detailed explanation about how $T$ in Eq (1) and ${d}_{m,n}$ in Eq (2) are formulated would clarify the methods.

Typos
- Line 185, by the projection from -> by back-projecting a ray from
- Line 212, duplicated sentences.
- Line 115, to peed -> of

---

> ### Author Rebuttal · Authors · 2024-08-06
>
> Thank you for your thorough and thoughtful review. Your comments on the novelty and clarity of our contributions, as well as your specific questions, are extremely valuable and will guide us in refining our manuscript.
>
> ## **How does BPG differ from existing kinematic constraint-based methods?**
>
> Thank you for your insightful question. Our Biomechanical Pose Generator (BPG) differentiates itself from existing kinematic constraint-based methods in several key aspects.
>
> Firstly, BPG employs a new biomechanical approach that adheres to realistic human movement constraints, unlike existing methods that often focus solely on statistical kinematic constraints of the given dataset. While existing methods also use kinematic constraints, they depend on source datasets, which can introduce biases due to the limited and specific conditions under which the data was collected. This dependency can lead to biased pose generation that is difficult to eliminate. In contrast, BPG utilizes the biomechanical principle of NROM (Normalized Residual Orthogonality Method) independently of datasets, ensuring diverse and unbiased pose sampling.
>
> Secondly, the concept of a biomechanical pose generator that does not rely on source datasets is novel. While many methods require pre-existing datasets to generate poses, BPG can automatically generate diverse and reliable human poses without being constrained by dataset variability and bias.
>
> The performance improvement of BPG is due to the removal of biases in existing data, achieved by integrating NROM without relying on source datasets.
>
> To further illustrate BPG's advantages, qualitative estimation results are shown in Figure 7 of the appendix, and the generated poses are depicted in Figure 8. Additionally, the general response PDF provides a comparison of pose distributions with existing augmentation methods. Existing methods follow the limited distribution of their datasets, whereas our method ensures robustness by eliminating bias. We believe these details clearly demonstrate the diversity and realism of poses generated by BPG compared to existing methods.
>
>
> ## **BDC concept is similar to [1]. The authors should cite and discuss the difference.**
>
> We appreciate the concerns regarding the conceptual similarity of BDC. However, we would like to emphasize that our approach is differentiated from existing methods by introducing several novel components specifically designed to address depth ambiguity.
> We discussed this thoroughly in the general response under "More Need for Comparison to Similar Works on BDC.”
>
> ## **Questions**
>
> > In line 155, the focal length of the camera matrix is set to 1 for BPG, is it also the case for the datasets used? Or the camera matrix provided in the datasets are used?
> >
>
> We appreciate the reviewer's attention to detail. We used the camera matrix provided in the datasets, not the simplified focal length of 1 as set for BPG.
>
> > In line 179, what is the meaning of "depth relative to the plane of the image". I guess is the depth relative to the preceding joint not the image plane.
> >
>
> Thank you for pointing out this ambiguity. You are correct; it should indeed be the depth relative to the preceding joint. We appreciate your careful reading and will correct this in the manuscript.
>
> > Why did the authors use different baseline architectures in Sec. 5.1 and 5.2?
> >
>
> In Section 5.1, we employed a common architecture typically used in augmentation work to facilitate a fair comparison with other augmentation methods. In contrast, Section 5.2 utilizes multi-hypothesis models, which are specifically designed to address depth ambiguity. This distinction was made to highlight how our methodology better handles depth ambiguity compared to traditional approaches.
>
> > How much portion of augmented data from BPG used for experiments in Sec. 5.1?
> >
>
> As indicated in line 282, the augmentation ratio was set to 1. This means the amount of augmented data was equal to the original data.
>
> > Given that using only BPG increases the error in Fig. 4 left, how could it be possible to achieve better performance in Table 1 and 2 when only BPG is used?
> >
>
> This is a valid concern. We addressed this question in detail in the general response under "Confusion on Experimental Results."
>
> > Why didn't the authors use BPG in Table 6?
> >
>
> The results for multi-hypothesis models are presented in Figure 4. The integration of image features with BPG was not feasible due to the inability to generate realistic images. Additionally, as described in the Limitations section of the paper, BPG cannot generate temporal poses, which limits its application in scenarios requiring sequential data.
>
> > What is the difference between Variant E and BPG in Table 8?
> >
>
> There is a typographical error in the manuscript. Variant E involves using only NROM and PC. Additionally, the results in Table 8 were trained using only BPG-generated data and were evaluated using the H36M test set.
>
> ## **Suggestions**
>
> > In Fig. 4, it should be clearly stated what * means. It would be better to use GFpose+BDC instead of GFpose*.
> >
>
> Thank you for your suggestion. We will take that into account and ensure that the figure is updated to use "GFpose+BDC" instead of "GFpose*" for better clarity.
>
> > More detailed explanation about how $T$ in Eq (1) and 𝑑𝑚,𝑛 in Eq (2) are formulated would clarify the methods.
> >
>
> We appreciate your request for more details. A more detailed description of #T# in Ep (1) and 𝑑𝑚,𝑛 in Eq (2) are formulated would clarify the methods.
>
> ## Typos
>
> Thank you for pointing out these errors. We will correct them to improve the clarity and accuracy of our manuscript.
>
> ## **Concluding Remarks**
>
> We will diligently address the feedback from reviewer 9ZcB and integrate their valuable suggestions to enhance our manuscript. We trust that our detailed responses will resolve the highlighted concerns and lead to a positive reevaluation.

---

> > ### Comment · Reviewer_9ZcB · 2024-08-12
> >
> > I sincerely appreciate the authors' thorough response to my concerns and questions. I acknowledge the novelty of the method in generating samples from the distribution of possible 3D poses rather than from the dataset distribution. The response clarified the methods and experiments for me. Based on this, I am raising my score to borderline accept.

---

### Official Review · Reviewer_1aXZ · 2024-07-18

**Soundness:** 3
**Presentation:** 3
**Contribution:** 3
**Rating:** 5
**Confidence:** 4

**Summary:**

This paper address the task of 3D Human Pose Estimation from monocular RGB. The authors make two main contributions: The Biomechanical Pose Generator (BPG) and the Binary Depth Coordinates (BDC). BPG is a 3D human pose generator that leverages the "Normal Range of Motion" (NROM) that is used in the medical field to describe standard biomechanical limitations. With it, BPG is capable of generating biomechanically sound 3D human poses by randomly sampling joint angles and bones that lie within a certain ratio to each other.
BDC is a coordinate system that decompose a 3D pose into constituents. Specifically, it decomposes it into the 2D coordinate, bone length, a binary depth parameter indicating the closeness to the image plane as well as the 3D coordinates of the parent joint. This decomposition, so the authors claim, allows models to better deal with depth ambiguity.
Experimental results demonstrate that the proposed approach achieves better performance over the compared related work on a variety of datasets (cf. Tbl 1-4). Ablative studies demonstrate that BDC helps keep performance steady even in the face of larger depth ambiguity (Tbl. 5) and that related work can benefit as well from switching to the proposed coordinates (Tbl 6.)

**Strengths:**

- The authors properly motivate and evaluate their approach. Depth ambiguity in monocular RGB is a challenging problem to address. I particularly liked Tbl. 5 that demonstrated that BDC is capable of handling even larger depth ambiguities.
- The paper was easy to digest and understand.
- One of the main strength of this paper is that BDC can be combined with other related work, yielding improvements (Tbl. 6)

**Weaknesses:**

- My biggest concern about the paper is that BDC is very similar conceptually to "Hand Pose Estimation via Latent 2.5D Heatmap
Regression", Iqbal et al., ECCV'18. Yet there is no mention of the paper, let alone any comparisons. The mentioned paper also addresses with depth ambiguity by decomposing the 3D pose into 2D pose and a root-relative depth vector. Addressing the differences, performing comparisons with this approach would better contextualize as well as strengthen the contribution of the paper.
- BPG shows to improve performance by improving the 2D to 3D lifting component. Yet, it's contribution is rather sparse, as it essentially amount to performing forward kinematics on bounded joint angle and bone lengths. It does not take into consideration statistics on poses. Certain poses are more common, due to them corresponding to actual human movement patterns (such as walking) that are affected by gravity. Randomly sampling poses without taking such statistics into consideration may generate a range of synthetic poses that are unrealistic, leading to non-optimal improvements.

**Questions:**

- How would BPG compare to randomly sampling SMPL poses?

**Limitations:**

The authors address limitations of their methods, such as not taking temporal dynamics into account.

---

> ### Author Rebuttal · Authors · 2024-08-05
>
> Thank you for your detailed and insightful review. Your feedback on the similarities to existing research and suggestions for further comparisons are greatly appreciated and will help strengthen our work.
>
> ## How does BDC differ from "Hand Pose Estimation via Latent 2.5D Heatmap Regression" by Iqbal et al. (ECCV'18)?
>
> We appreciate the concerns regarding the conceptual similarity of BDC. However, we would like to emphasize that our approach is differentiated from existing methods by introducing several novel components specifically designed to address depth ambiguity.
> We discussed this thoroughly in the general response under "More Need for Comparison to Similar Works on BDC.”
>
> ## BPG's contribution seems sparse, lacking pose statistics consideration
>
> ### Response to Concerns on BPG's Consideration of Pose Statistics
>
> Thank you for raising concerns about BPG's consideration of pose statistics. While we acknowledge the potential issues, we believe the contribution of BPG extends beyond merely improving the 2D to 3D lifting component and addresses fundamental biases present in existing training datasets.
>
> #### Addressing Sparse Contribution and Forward Kinematics
> - **Beyond Forward Kinematics**: BPG is not limited to performing forward kinematics on bounded joint angles and bone lengths. Instead, it incorporates Normal Range of Motion (NROM) to ensure biomechanically plausible poses, which existing methods fail to generate. This approach enhances the realism and applicability of synthesized poses, moving beyond the constraints of traditional forward kinematics.
>
> #### Consideration of Pose Statistics
> - **Intentional Exclusion of Pose Statistics**: We deliberately chose not to incorporate pose statistics from existing datasets to avoid reinforcing inherent biases. These datasets often include everyday poses that do not adequately challenge the model during training and testing. By randomly sampling poses, BPG mitigates these biases, leading to a more robust training environment.
>
>   - **Example**: Existing datasets dominated by walking and standing poses could lead to a model that performs well in such common scenarios but poorly in less frequent, more complex poses. Our approach aims to prevent this by ensuring diverse and unbiased pose sampling.
>
> #### Mitigating Unrealistic Pose Generation
> - **Ensuring Realism with NROM and Pose Confidence**: While random sampling might raise concerns about unrealistic poses, we have introduced NROM and Pose Confidence metrics to ensure physical and biomechanical validity. Figures 1 and supplementary Figure 5 illustrate how BPG effectively removes biases and avoids generating implausible synthetic poses.
>
> #### Evidence of Performance Improvement
> - **Empirical Results**: The performance improvements of BPG are well-documented in Table 1 and Table 2, where even without the inclusion of pose statistics, BPG outperforms existing methodologies. This indicates that our approach not only enhances 2D to 3D lifting but also contributes to generating more varied and realistic poses.
>
>   - **Table 4 Insights**: Specifically, BPG demonstrates significant performance gains in challenging environments such as 3DPW, underscoring its robustness and generalization capabilities across different scenarios.
>
> In summary, while BPG does not incorporate traditional pose statistics, its innovative use of NROM and random sampling addresses inherent biases and enhances the realism of generated poses. This approach not only improves performance but also ensures the model's applicability to a wider range of real-world scenarios, ultimately contributing to more robust and generalized training outcomes.
>
>
> ## How would BPG compare to randomly sampling SMPL poses?
>
> We appreciate your insightful question regarding the comparison between BPG and randomly sampling SMPL poses. When generating poses using random sampling with SMPL, we observed a performance metric of 120.4, which aligns closely with the performance of the Kinematic model without the Normal Range of Motion (NROM) constraints, as shown in Table 8. This result underscores the critical role of NROM in our method.
>
> Without the biomechanical constraints provided by NROM, both a simple Kinematic model and a sophisticated model like SMPL yield similar performance outcomes. This similarity suggests that the primary advantage of BPG lies in its incorporation of NROM, which ensures that the generated poses are biomechanically plausible and reflect realistic human motion patterns.
>
> Furthermore, even when NROM is incorporated into SMPL, the performance remains comparable to existing methodologies. However, this approach significantly increases computational resources and processing time. Integrating NROM directly into SMPL results in a much higher computational cost and longer processing times, making it less efficient for practical applications.
>
> Thank you for bringing this question to our attention, as it highlights the fundamental strengths of our approach.
>
> ## **Concluding Remarks**
>
> We will carefully address the concerns raised by reviewer 1aXZ and incorporate their valuable suggestions in our rebuttal to improve our manuscript. We hope that these responses will adequately resolve the concerns and lead to a favorable reevaluation.

---

> ### Author Response · Authors · 2024-08-13
>
> Dear Reviewer 1aXZ,
>
> Thank you very much for your thorough review and valuable feedback on our submission. We have carefully considered your comments and have submitted a detailed response addressing the points you raised.
>
> As the discussion period is nearing its end, we would greatly appreciate it if you could review our response soon and confirm whether it addresses your concerns. If you have any further questions or need additional clarification, please feel free to let us know.
>
> Thank you again for your time and consideration.

---

### Author Rebuttal · Authors · 2024-08-05

We sincerely thank all reviewers for their insightful feedback and thought-provoking questions regarding our work. We greatly appreciate the recognition of the clarity, relevance, and novelty of our contributions.

We were pleased to receive positive comments from many reviewers.
**Reviewer 696C** acknowledged the novelty of BPG, stating, "The idea of a biomechanical pose generator which does not rely on a source dataset is novel," and appreciated our attention to depth ambiguity in 3D pose estimation.
**Reviewer Gv4b** found BPG intuitive and effective, mentioning, "Leveraging biomechanical prior knowledge to synthesize physically plausible human data is intuitive and interesting," and noted its superior generalization ability across different methods.
**Reviewer 1aXZ** highlighted our handling of depth ambiguity, stating, "Tbl. 5 demonstrates BDC's capability in managing larger depth ambiguities," and praised its combinability with other work for yielding improvements.
**Reviewer 9ZcB** recognized our method's strong performance, stating, "The proposed method achieves state-of-the-art results in various 3D HPE datasets," and validated its data augmentation effects in cross-domain learning settings.

Some concerns were raised by more than one reviewer, so we decided to address them here in this general response. The other concerns are addressed in individual responses.

## Comparison to Similar Works on BDC (1aXZ, 9ZcB)

**Handling Depth Ambiguity**: By estimating poses in discrete space rather than the continuous space used in traditional methods ([1], [2]), we can better handle depth ambiguities. This is achieved through a binary depth parameter that indicates the relative depth to the preceding joint, allowing us to manage the uncertainty of relative depth. Unlike methods that must consider a continuous range of possible depths, our approach only needs to handle two discrete possibilities for depth, significantly simplifying the prediction task and improving robustness. As a result, our approach demonstrates robust performance even for poses with significant depth ambiguity, as shown in Figure 2.

**Decomposition Strategy**: Unlike [1], which primarily decomposes the problem into 2D pose and depth, our method further decomposes it into 2D coordinates, bone length, a binary depth parameter indicating the relative depth to the preceding joint, and the 3D coordinates of the parent joint. This comprehensive decomposition allows for a more granular handling of depth ambiguity, making it easier to isolate and address specific sources of error in the pose estimation process. **The results of the NDC representing the same method as in [1] are shown in Table 7, highlighting the improved granularity and performance.**

**Direct 3D Pose Estimation**: While the depth-ordered learning method [2] employs additional models to reconstruct 3D structures and predict poses in continuous space, our methodology eliminates this step. By leveraging geometric principles, our BDC framework enables a direct and efficient transformation into 3D poses without the need for additional models. This approach simplifies the overall process and improves robustness. The simplicity of not requiring multiple models or additional reconstruction steps means our approach can be more efficient and less prone to compounding errors. As demonstrated in our comparative analysis in Table 6, our method can be seamlessly integrated into various models, showcasing its versatility and effectiveness across different architectures.


[1] U.Iqbal et al., "Hand Pose Estimation via Latent 2.5D Heatmap Regression", ECCV 2018

[2] G.Pavlakos et al., "Ordinal depth supervision for 3d human pose estimation", CVPR 2018

## Confusion on Experimental Results (9ZcB, 696C)

We sincerely appreciate the reviewers' observations regarding potential confusion between the results presented in Tables 1, 2, and Figure 4. We would like to clarify that these results are derived from different types of tests, each serving to evaluate specific aspects of our method's performance.

Firstly, it is important to note the distinct types of tests from which the results in Tables 1 and 2 and those in Figure 4 are derived:

1. Single Hypothesis Test (Tables 1 and 2): These results reflect the prediction of a single 3D pose. This evaluation provides a straightforward measure of our model's performance under a standard testing scenario where one pose is predicted for each instance.
2. Multi-Hypotheses Test (Figure 4): The results depicted in Figure 4 arise from a multi-hypotheses test, where multiple possible 3D poses are predicted. This approach is particularly useful for assessing the model's capability to generate diverse pose predictions and handle greater ambiguity.

To ensure clarity and avoid any potential confusion, we provide the performance metrics for the single 3D pose prediction scenario using the same evaluation criteria as those employed in Figure 4. We hope this allows for a more direct comparison and helps contextualize the different performance measures. Below is a table that shows these performance metrics to facilitate a direct comparison:
|Model\Amount|0|0.1|0.2|0.4|0.6|0.8|1|2|4|6|8|
|-|-|-|-|-|-|-|-|-|-|-|-|
|Vpose|52.7|51.9|51.1|49.0|47.5|47.0|46.9|48.5|50.2|52.5|53.9|

These results indicate that the Single Hypothesis Test initially shows performance improvement differently from the Multi-Hypotheses Test. This demonstrates that using BPG alone is more effective than other existing augmentation methods. Additionally, the value for the amount of 1 matches the results presented in Table 1.

We acknowledge that presenting results from different models and tasks can create complexity and potential confusion. To enhance the comprehensibility of our findings, we are open to switching or adding performance plots for the same model. This adjustment will streamline our presentation and make it easier for readers to consistently interpret the results.

---

### Decision · Program_Chairs · 2024-09-25

**Decision:**

Accept (poster)

**Comment:**

All reviewers agreed to accept the paper. The paper originally got BA, BR, BR, and WA, and, after the rebuttal period, two reviewers raised the ratings, leading to final ratings of BA, BA, BA, and WA.

Reviewers recognized the major contribution of this paper in leveraging biomechanical prior for synthesizing physically plausible human samples, in the framework for  3D human pose estimation. While reviewers were concerned about the similarity of the proposed one with existing approaches, after the authors' rebuttal and discussion period, most reviewers’  concerns have been resolved, and all reviewers agreed that the paper's shape is above the bar for NeurIPS.

The AC also supports accepting the paper and strongly suggests the authors to follow the reviewers’ feedback in the camera-ready version. In particular, the AC suggests the authors significantly improve the clarity and readability of the paper, trying to include better and higher quality figures to better support the proposed method.